# Determinants of Residents' Support for Sustainable Tourism Development: An Empirical Study in Midyat, Turkey

Abdullah Uslu [1], Emrullah Erul [2], José António C. Santos [3,*], Sanja Obradović [4] and Margarida Custódio Santos [3]

1    Department of Tourism Management, Manavgat Tourism Faculty, Akdeniz University, 07600 Antalya, Turkey; auslu@akdeniz.edu.tr

2    Department of Tourism Management, Tourism Faculty, İzmir Katip Celebi University, 35620 İzmir, Turkey; emrullah.erul@ikcu.edu.tr

3    School of Management, Hospitality and Tourism, Campus da Penha, Universidade do Algarve, 8005-139 Faro, Portugal; mmsantos@ualg.pt

4    Department of Geography, Tourism and Hotel Management, Faculty of Science, University of Novi Sad, 21000 Novi Sad, Serbia; sanjaobradovic992@gmail.com

*    Correspondence: jasantos@ualg.pt

**Abstract:** The purpose of this study is to examine the extent to which residents' perceptions of their place image can predict their perceptions of the impacts of tourism, and ultimately, to elucidate their support for sustainable tourism development in Midyat, Turkey. This city currently faces a range of negative impacts associated with tourism, such as inflation, high leakage, threats to family structures, environmental degradation, pollution, and crowding. At the same time, Midyat seeks to maximize the positive impacts of tourism, including job creation and employment, environmental protection, cultural and educational enrichment, and recreational opportunities. Therefore, there is an urgent need for a strategic shift in development. The study population consists of residents residing in Midyat, Turkey, from whom three hundred and fifty-six questionnaires were collected. PLS (Partial Least Squares) path analysis was utilized to analyze the research model constructed based on the literature. The results demonstrated that residents' place image significantly predicted two out of three impacts of tourism, namely, socio-cultural and economic impacts. Additionally, residents' perceptions of environmental and socio-cultural impacts were significant predictors of their support or opposition to sustainable tourism. While perceived environmental impacts have a positive effect on support for sustainable tourism, perceived economic impacts have a negative effect. This finding can guide tourism planners and professionals to make more informed decisions and take stronger steps toward sustainable tourism development. This study revealed that the quality of life, job opportunities, infrastructure, and economic income of Midyat's residents are insufficient. However, the city is characterized by social attributes, such as friendliness, trust, and supportiveness. The results are limited to local residents, and using only one tourist stakeholder to assess sustainable tourism development is insufficient. Therefore, additional research must be performed to guarantee the involvement of other tourism stakeholders.

**Keywords:** perceived tourism impacts; residents' place image; sustainable tourism development; Midyat

## 1. Introduction

In the past 60 years, the tourism industry has experienced rapid growth. Tourism authorities and scholars expected this trend to continue until the outbreak of COVID-19, which has had a negative impact on the industry globally [1–6]. Urban tourism has been one of the most dynamic market segments in the tourism sector, as it generates revenue that promotes destination development [7–9]. Recently, city breaks have emerged as one of the fastest-growing tourism segments in Europe, with significant growth rates [10,11].

However, the sustainability of city spaces raises questions regarding their ability to differentiate themselves territorially, particularly in terms of tourism practices, in addition to their potential for transforming urban areas [12].

The development of urban tourism has the potential to disrupt the lives of residents [13], although it can also have multiple positive impacts on local communities [14,15]. For instance, it can enhance the quality of life of residents by creating employment opportunities and improving infrastructure and amenities [16–18]. Sustainable tourism development can also stimulate the economy [19], promote reputation and competitiveness of tourism destinations [20–22], and preserve resources and the environment [23–25]. However, there are also negative impacts associated with tourism [26], as is the case of urban tourism. For example, urban populations may experience overtourism, traffic congestion, increased crime rates, drug use, higher living costs, and cultural changes resulting from tourism activities [27,28]. Additionally, tourism development can lead to the degradation of natural resources [29].

Tourism stakeholders, particularly residents, carefully consider the impacts of tourism on their lives and the environment [30]. For instance, residents may oppose further tourism development if they are excluded from the planning process [31], or if the negative impacts of tourism outweigh the positive ones [13,16]. Therefore, scholars have studied residents' perceptions of tourism and their attitudes toward tourism development, as tourism impacts can significantly affect residents' daily lives and influence their support for sustainable tourism [1,32–34].

Previous studies have investigated variables that predict residents' support for tourism development. Some scholars have focused on residents' perceived impacts of tourism [34–38] and residents' place image [35,39,40], while others have examined residents' perceptions of sustainable tourism development [33,41,42]. However, there is still a need to examine these variables together. Therefore, this study aims to fill this research gap by examining how residents' place image predicts their perceptions of tourism impacts and, ultimately, their support for sustainable tourism development in Midyat, Turkey.

## 2. The Literature Review

### 2.1. Urban Sustainable Tourism Development and City Image

In recent decades, urban policies in many cities have incorporated sustainable development strategies, reflecting a new approach to urban development [43,44]. Sustainable development seeks to satisfy the needs and demands of present generations without jeopardizing the ability to meet those of future generations [45]. Sustainable urban development strives to balance the socio-cultural, environmental, and economic dimensions of the urban environment [46]. The concept of a "sustainable city" encompasses the restoration of public spaces, the preservation of natural areas in and around the city, and the adoption of sustainable transportation options [47].

Furthermore, sustainable development is an essential part of community building [48] and focuses on the environment, social development, economic development [34,49], and the preservation of identity, art, and heritage [48]. Urban tourism is a specific form of tourism that takes place in a city [11], and urban tourists' top motivation is the cultural journey that involves discovering a city's tangible and intangible heritage [14]. According to Kalaivani et al. (2018) [50], there must be a link between urban planning and urban tourism, which is crucial for creating a friendly and peaceful relationship between tourists and residents.

Organisations, such as UNESCO (2017), have reinforced the focus on sustainable building practices in architecture and design education, as well as urban planning, and their actions have raised awareness of sustainable development and the importance of guiding future professionals on how to create sustainable cities [51–53]. Urban destinations include cities, towns, and various features of the built environment [54]. Cities represent entertainment centres, historical places, art, food, gateways to traditional and modern society, and central points for trade, finance, and industry. Furthermore, cities must

offer valuable attractions to tourists, such as unique activities, heritage, nature, food, and events [55].

New opportunities and innovative solutions for sustainable urban tourism can be achieved through the cooperation, knowledge, expertise, and capital resources of all stakeholders involved. For development to be sustainable, tourism must consider the daily lives of local people, the preservation of local activities and public spaces, the harmonious integration of accommodation facilities in the city, and the adoption of sustainable transportation mobility [54]. The residents' perception of the place image can significantly influence their acceptance or rejection of further tourism development [34,56].

Stylidis et al. (2014) [35] examined the place image as a crucial factor in understanding residents' perceptions of tourism impacts. Furthermore, Stylidis (2016) [40] developed an original model to test the relationship between sub-dimensions of residents' place image (e.g., physical appearance, entertainment services, community services, and social environment) and their perceived tourism impacts. Stylidis (2016) [40] found support for two out of the four dimensions of residents' place image (i.e., social environment and physical appearance) in relation to their perceived tourism impacts. Based on the results in the literature, the following hypotheses have been formulated:

**H1.** *Residents' perceptions of the sociocultural impacts of tourism positively influence their place image.*

**H2.** *Residents' perceptions of the economic impacts of tourism positively influence their place image.*

**H3.** *Residents' perceptions of the environmental impacts of tourism positively influence their place image.*

### 2.2. Residents' Perception of Tourism Development Impacts

Researchers have often discussed residents' supportive attitudes regarding tourism impacts [27,37], as residents' attitudes towards tourism development and their perceptions of tourism impacts are crucial for sustainable tourism development [1,57]. Researchers who have studied residents' support for tourism have classified the impacts of tourism into three dimensions: socio-cultural, environmental, and economic [36,58–67]. Therefore, tourism sustainability must encompass the triple bottom line, addressing those three dimensions [68]. A better understanding of the impacts of tourism development can be ensured by appropriately managing tourism benefits and costs for residents. Economic impacts of tourism relate to elements, such as increased prices, employment, revenue, and infrastructure development [69]. Socio-cultural impacts of tourism have a strong effect on residents' social and cultural life, including the quality of life, service quality, events, traditions, cultural heritage, traffic, and crime [70]. The environmental impacts of tourism affect the environment, including air, soil, water, and noise pollution, land construction, depletion of natural resources, and litter production [71,72]. Tourism planners should encourage residents to participate in the process of tourism development and keep them informed of all possible impacts of tourism development. If residents are involved and well informed, they will support the sustainable development of tourism [16,31].

### 2.3. Support for Tourism Development

Without a doubt, tourism development has evident impacts on local communities, both positive and negative. The literature has confirmed that the support of residents is crucial for the successful development of tourism [1,35,73]. Previous studies [31,74,75] have confirmed that resident involvement is critical to the sustainable planning and development of tourism. For instance, Erul et al. (2022) [74] found that residents' intentional support for tourism and pro-tourism behavior had significant effects on involvement in tourism.

According to Sharpley (2014) [76], perceived positive impacts of tourism encourage residents' support for sustainable tourism development. Conversely, perceived negative effects of tourism can adversely affect residents' support for tourism development [37,77,78].

It can be concluded that residents' support for tourism may vary, depending on the impacts of tourism. In other words, residents will be willing to support tourism development if their perception is positive or if they perceive the benefits of tourism (i.e., positive impacts of tourism). Following the literature review, hypotheses H4, H5, and H6 have been formulated as follows:

**H4.** *Residents' perceptions of perceived socio-cultural impacts of tourism positively influence residents' support for tourism development.*

**H5.** *Residents' perceptions of perceived economic impacts of tourism positively influence residents' support for tourism development.*

**H6.** *Residents' perceptions of perceived environmental impacts of tourism positively influence residents' support for tourism development.*

### 3. Materials and Methods

*3.1. Study Method*

The research design of the empirical study was developed based on a causal approach. The study model, depicted in Figure 1, was constructed in accordance with the hypotheses derived from the literature review.

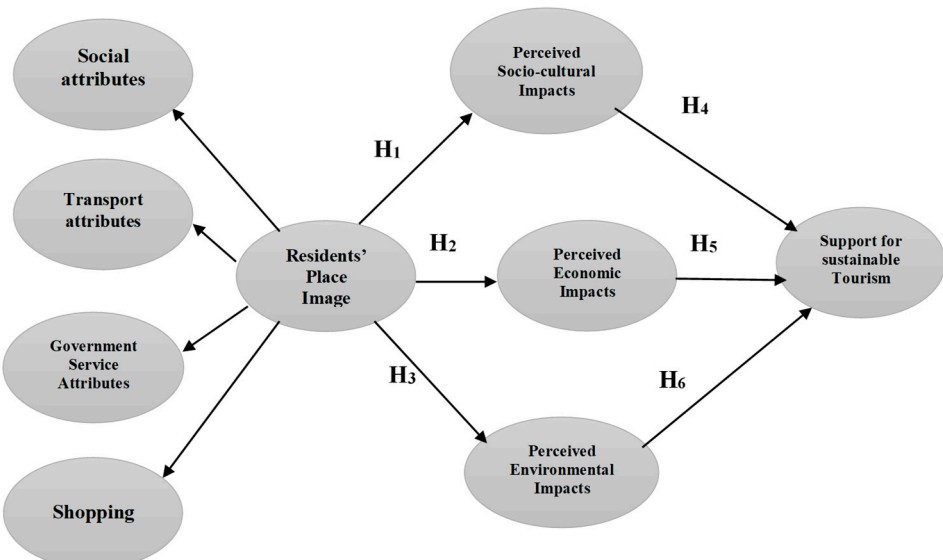

**Figure 1.** The proposed research model.

*3.2. Study Instrument*

This study aims to investigate the relationship between the place image of Midyat residents and their perceptions of the impacts of tourism, specifically in terms of socio-cultural, economic, and environmental dimensions, as well as how this ultimately affects their support for tourism development. The Midyat residents' place image comprises 14 items, which are divided into four sub-dimensions: government service attributes, transportation attributes, shopping, and social attributes. The scale for support of urban tourism development, which is unidimensional, consists of three items and was adapted from Ramkissoon and Nunkoo's (2011) [37] research. Thirteen statements that pertain to the socio-cultural (four items), perceived economic (five items), and environmental (four items) impacts were adapted from previous studies by Nunkoo and Ramkissoon (2010) [27], Stylidis et al. (2014) [35], and Gursoy and Rutherford (2004) [79].

### *3.3. Study Location*

Midyat is located in Mesopotamia, which is considered the oldest settlement area in the world. Throughout history, this area has witnessed the dominance of many civilizations, such as the Sumerians, Assyrians, Urartians, Macedonians, Persians, and Romans. Later on, the Muslim states of the Abbasids, Umayyads, Artuqids, Seljuks, and Ottomans became dominant. Due to the influence of different religions and sects, such as Islam, Christianity, and Yezidi, Turkish became a meeting point for Kurdish, Arabic, and Syriac cultures. Midyat's historic houses, streets, and avenues offer visitors many places to explore, including the Cevat Pasa and Haci Abdurrahman mosques, Ulu Mosque, Deyr-ul Umur, and Mor Gabriel monasteries, Mor Smuni, Mor Barsavmo, Mor Aksanoya, Mor Sabrel, Virgin Mary, Mor Abraham, and Protestant churches, as well as the cave museum and State Guest House [80].

The positive development of Midyat's city image as a center for history, culture, and religious tourism is due to the large-scale production of movies (such as Hükümet Kadın 1 and 2) and TV series on international and national channels (such as Berivan, Aşka Sürgün, Sıla, Adını Kalbime Yazdım, and Hercai), which have made it a popular destination. As a result, Midyat received over 1 million tourists in the first eight months of 2018 [81].

### *3.4. Study Sampling*

According to the 2020 data from the Turkish Statistical Institute (TSI), the population of Midyat is 117,364 inhabitants. Due to the impossibility of reaching all residents in Midyat, the questionnaires were collected by using convenience sampling technique between May and August, 2020. While a total of 385 questionnaires were obtained, 29 questionnaires (due to deficiencies or errors) were excluded from the study. As a result, the study was conducted with the remaining 356 questionnaires. According to Bryman and Cramer (2001) [82], a sufficient sample size can be obtained by multiplying the number of statements by five or ten. Therefore, in the questionnaire designed for this study, which includes a total of 30 statements, a minimum acceptable sample size of $30 \times 100 = 300$ was determined for the questionnaire population. Thus, it can be concluded that the 356 questionnaires obtained are sufficient to represent the population [82].

### *3.5. Data Collection and Analysis*

The questionnaire comprises two parts. The first part includes eight questions regarding the demographic characteristics of the residents, such as gender, age, level of education, marital status, monthly income, occupational status, birthplace, and length of residency in Midyat. The second part consists of 30 items that measure the variables of the study. Participants were asked to respond using a 5-point Likert scale (5 = Strongly Agree . . . 1 = Strongly Disagree). Three social attributes were rated on a 1–5 bipolar scale representing contrasting views, where 5 = friendly, supportive, and trusting, and 1 = unfriendly, distrusting, and hostile. Data analysis was performed using SPSS 22.0 and SmartPLS 3.3.0 statistical software.

## 4. Findings

In the findings section, the study presents the residents' demographic characteristics, the validity and reliability analyses of the scale, the measurement model (outer model), descriptive statistics, correlation findings, and hypotheses formulated with the structural equation model (inner model).

### *4.1. Demographic Characteristics of the Sample Population*

Table 1 shows that the majority of the sample population are male (58.7%), married (54.2%), young (between the ages of 25–34), white-collar workers (60.6%), and highly educated (63.5%). When their monthly income status is examined, it has been determined that 28.7% earn a monthly income between 4000–5000 TL.

**Table 1.** Demographic characteristics of the sample.

| Variables | Percentage (%) | Variables | Percentage (%) |
|---|---|---|---|
| | (*n* = 356) | | (*n* = 356) |
| Gender (%) | | Occupation (%) | |
| Male | 58.7 | Student | 7.6 |
| Female | 41.3 | White collar worker | 60.6 |
| Age Group (%) | | Blue collar worker | 7.3 |
| 18–24 age | 11.8 | Own business | 6.5 |
| 25–34 age | 55.6 | Other | 18.0 |
| 35–44 age | 26.1 | Length of residency (%) | |
| 45–54 age | 4.2 | Less than 1 year | 10.7 |
| 55–64 age | 2.2 | 1–5 | 20.2 |
| Level of Education (%) | | 6–10 | 11.2 |
| Primary school | 4.5 | 11–15 | 4.8 |
| High school | 13.2 | 16–20 | 9.0 |
| Associate degree | 7.3 | 21 years and more | 44.1 |
| University graduate | 63.5 | Monthly Income (TL) (%) | |
| Master's or Ph.D. | 11.1 | No income | 18.5 |
| Marital status (%) | | Minimum wage | 5.3 |
| Married | 54.2 | TL 3000–4000 | 11.5 |
| Single | 45.8 | TL 4000–5000 | 28.7 |
| Place of birth (%) | | TL 5000–6000 | 22.5 |
| Midyat | 60.1 | TL 6001 and more | 13.5 |
| Mardin | 12.9 | | |
| Other places | 27.0 | | |

*4.2. Validity and Reliability Analysis of the Scale and the Measurement Model (Outer Model)*

Prior to analyzing the research model, validity and reliability studies were conducted on the variables included in the study using the measurement model. The PLS-SEM method was used for confirmatory factor analysis (CFA) on the measurement model [83]. As part of the measurement model testing, internal reliability, convergent validity, and divergent validity were evaluated. To assess the overall reliability of the study, Cronbach's Alpha ($\alpha$) was examined, while Composite Reliability (CR) values were tested for internal reliability. Convergent validity was determined using Average Variance Extracted (AVE) values. In order to meet the criteria for good reliability and validity, factor loadings need to be above 0.60, CR value and Cronbach's Alpha ($\alpha$) should exceed 0.70, and AVE values need to be over 0.50 [84]. Table 2 presents information on the Cronbach's Alpha ($\alpha$), convergent validity, and internal consistency reliability of the structures used in the study.

After evaluating the internal reliability, the convergent and discriminant validities of the measurement model are assessed. The convergent validity is determined by examining the standardized factor loadings, t-values, and their significance, as well as the average variance extracted (AVE) values. Information on the standardized factor loadings, t-values, and AVE values is presented in Table 2. The table shows that the standardized factor loadings are well over 0.70, t-values are well over 2.56 [85], and all significant dimensions have AVE values over 0.50, except for the residents' place image dimension, which has an AVE value over 0.40. The lower AVE of the Midyat residents' place image could be due to the use of composite variables to represent this construct [84]. Furthermore, since the residents' place image is transformed into a secondary level structure, a low AVE value is expected. Therefore, it can be concluded that convergent validity is established. The measurement model comprises eight latent variables, consisting of 30 observed variables. The Midyat residents' place image dimension and the secondary level latent structure, which includes transportation attributes, social attributes, government service attributes, and shopping, are combined into a latent structure.

**Table 2.** Descriptives of the measurement items, standardized factor loadings, t-values, measurement model, and reliability coefficients.

| Dimensions and Items (*n* = 356) | Standardized Factor Loading | t-Values | Mean | Standard Deviation |
|---|---|---|---|---|
| *First Order* | | | | |
| Transport Attributes (Cronbach's Alpha = 0.833; CR = 0.90; AVE = 0.75) | | | | |
| Midyat's road connections are sufficient. [a] | 0.883 | 47.875 | 2.15 | 1.326 |
| Midyat's traffic is free/easy. [a] | 0.851 | 45.249 | 2.55 | 1.353 |
| Midyat's roads are well maintained. [a] | 0.863 | 44.130 | 2.11 | 1.249 |
| Social Attributes (Cronbach's Alpha = 0.823; CR = 0.89; AVE = 0.74) | | | | |
| Unfriendly-Friendly * | 0.845 | 37.012 | 4.20 | 1.067 |
| Distrusting-Trusting * | 0.881 | 49.814 | 4.22 | 1.108 |
| Hostile-Supportive * | 0.851 | 31.022 | 4.46 | 0.889 |
| Government Service Attributes (Cronbach's Alpha = 0.842; CR = 0.89; AVE = 0.68) | | | | |
| There are enough business areas in Midyat. [a] | 0.764 | 28.359 | 2.28 | 1.169 |
| One can trust the municipality because it makes healthy decisions. [a] | 0.888 | 60.226 | 2.33 | 1.197 |
| I am pleased with the services of the municipality oriented towards houses in Midyat. [a] | 0.829 | 37.358 | 2.38 | 1.287 |
| Public transportation is sufficient in Midyat. [a] | 0.813 | 38.955 | 2.40 | 1.323 |
| Shopping (Cronbach's Alpha = 0.840; CR = 0.893; AVE = 0.676) | | | | |
| There is a wide shopping variety in Midyat. [a] | 0.816 | 41.440 | 2.34 | 1.249 |
| One can find good quality home utilities stores. [a] | 0.858 | 50.630 | 2.63 | 1.211 |
| There are different types of restaurants in Midyat. [a] | 0.842 | 42.852 | 2.72 | 1.242 |
| There are markets in eligible locations in Midyat. [a] | 0.768 | 33.053 | 3.36 | 1.306 |
| Perceived Socio-Cultural Impacts (Cronbach's Alpha = 0.784; CR = 0.86; AVE = 0.61) | | | | |
| The recreation areas/places are sufficient in Midyat. [a] | 0.755 | 20.014 | 2.00 | 1.153 |
| The cultural activities/entertainment are sufficient in Midyat. [a] | 0.845 | 40.035 | 1.95 | 1.071 |
| One has the chance to meet people from different cultures in Midyat. [a] | 0.749 | 26.519 | 3.48 | 1.350 |
| There is a community atmosphere in Midyat. [a] | 0.765 | 29.496 | 3.35 | 1.335 |
| Perceived Economic Impacts (Cronbach's Alpha = 0.895; CR = 0.92; AVE = 0.71) | | | | |
| The living standard is sufficient in Midyat. [a] | 0.827 | 41.693 | 2.46 | 1.005 |
| Job opportunities are sufficient in Midyat. [a] | 0.897 | 66.454 | 2.05 | 0.983 |
| The infrastructure is sufficient in Midyat. [a] | 0.885 | 58.825 | 2.00 | 1.095 |
| The economic income of Midyat is sufficient. [a] | 0.884 | 53.875 | 2.16 | 1.092 |
| Land property/house prices of Midyat are reasonable. [a] | 0.703 | 17.935 | 1.86 | 1.138 |
| Perceived Environmental Impacts (Cronbach's Alpha = 0.706; CR = 0.78; AVE = 0.48) | | | | |
| Midyat is crowded. [a] | 0.722 | 4.012 | 2.91 | 1.112 |
| Traffic congestions occur in Midyat. [a] | 0.715 | 6.036 | 2.63 | 1.218 |
| Noise levels of Midyat are high. [a] | 0.650 | 4.170 | 2.53 | 1.152 |
| There is environmental pollution in Midyat. [a] | 0.671 | 4.824 | 3.10 | 1.337 |
| Support for Sustainable Tourism (Cronbach's Alpha = 0.870; CR = 0.92; AVE = 0.79) | | | | |
| Tourism must be developed, focusing on cultural and historical attractions. [a] | 0.910 | 32.780 | 4.36 | 0.910 |
| I am happy and proud that there are tourists coming to see what the city has to offer. [a] | 0.849 | 43.506 | 4.14 | 1.048 |
| I would want to see more tourism development in the city. [a] | 0.914 | 65.228 | 4.47 | 0.834 |
| *Second Order* | | | | |
| Residents' Place Image (Cronbach's Alpha = 0.877; CR = 0.90; AVE = 0.40) | | | | |
| Social Attributes | 0.486 | 10.815 | | |
| Transport Attributes | 0.738 | 22.881 | | |
| Government Service Attributes | 0.882 | 63.400 | | |
| Shopping | 0.794 | 33.340 | | |

[a]: A 5-point Likert scale was used for measurement, with 1 = strongly disagree . . . 5 = strongly agree. *: A bipolar scale was utilized, where 1 = unfriendly, distrusting, hostile and 5 = friendly, trusting, supportive. Note: The significance level of all factor loadings was *p* < 0.001, and bootstrapping was performed over 2000 samples.

Since the Cronbach's Alpha ($\alpha$) values range from 0.895 to 0.706 and the CR values range from 0.92 to 0.78 for all variables used in the study, it can be concluded that internal consistency reliability has been established. Moreover, the factor loads for all variables range between 0.862 and 0.668, and the AVE values are generally above 0.50, indicating

that convergent validity has been established. To determine divergent validity, the Fornell and Larcker Criterion was used [86,87]. According to Fornell and Larcker (1981), the square root of the AVE values in the study should be greater than the correlations between the other constructs in the study [88]. The results obtained using the Fornell and Larcker criterion (Table 3) demonstrate that divergent validity has been established. The correlation values range between the optimal values of ±0.3 and ±0.9 [89], indicating a significant and unidirectional relationship without any linearity problems (≥0.90).

**Table 3.** Results of discriminant validity.

| | Government Service Attributes | Perceived Economic Impacts | Perceived Environmental Impacts s | Perceived Socio-Cultural Impacts | Shopping | Social Attributes | Support for Sustainable Tourism | Transport Attributes |
|---|---|---|---|---|---|---|---|---|
| Government Service Attributes (RPI) | **0.824** | | | | | | | |
| Perceived Economic Impacts | 0.589 | **0.842** | | | | | | |
| Perceived Environmental Impacts | 0.164 | 0.190 | **0.690** | | | | | |
| Perceived Socio-cultural Impacts | 0.563 | 0.660 | 0.305 | **0.780** | | | | |
| Shopping (RPI) | 0.571 | 0.524 | 0.217 | 0.562 | **0.822** | | | |
| Social Attributes (RPI) | 0.262 | 0.231 | 0.147 | 0.319 | 0.285 | **0.859** | | |
| Support for Sustainable Tourism | 0.204 | 0.069 | 0.327 | 0.244 | 0.249 | 0.176 | **0.891** | |
| Transport Attributes (RPI) | 0.606 | 0.507 | −0.043 | 0.397 | 0.353 | 0.232 | 0.161 | **0.866** |

Notes: RPI = Residents Place Image, bolded elements are the square root of AVE. Elements below the AVE line are the correlations between the factors.

### 4.3. Testing the Structural Model (Inner Model)

In order to test the hypotheses of the study, the Structural Equation Model (SEM), shown in Figure 2, was used to determine if it was supported by the data. Partial Least Squares Structural Equation Modeling (PLS-SEM) was employed to analyze the model. Bootstrapping analysis was used to calculate the t-values used in the evaluation of the path (β) coefficients of PLS. This involved taking 2000 sub-samples from the main sample.

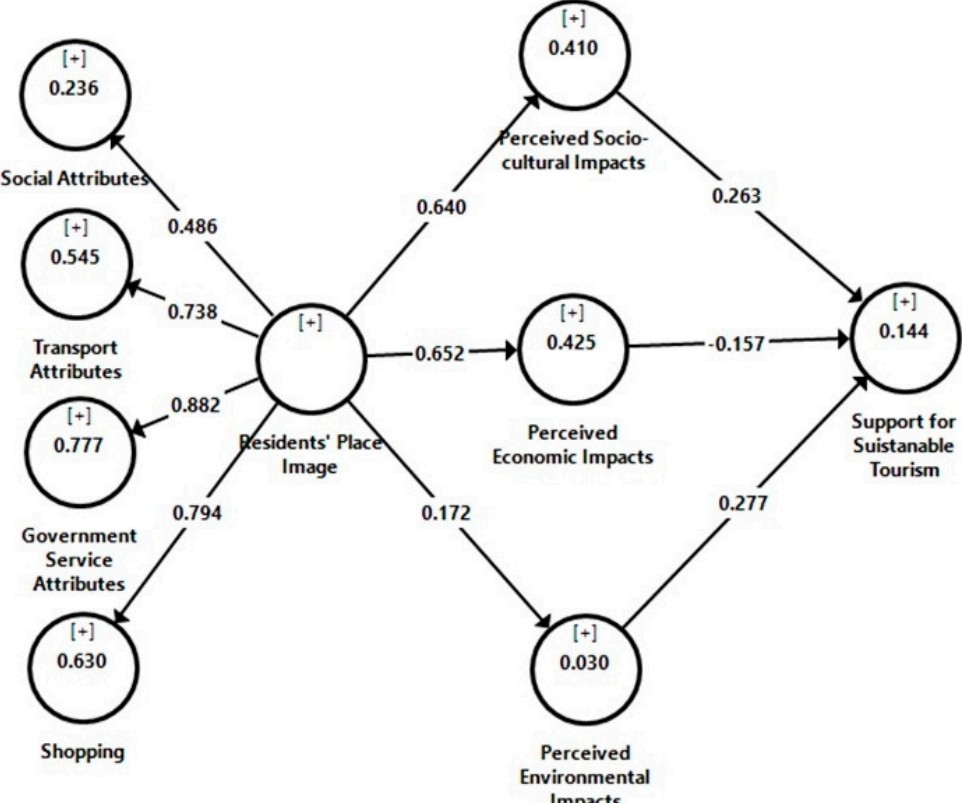

**Figure 2.** Results of regression paths and R2 values.

The results of the hypotheses formed using the proposed research model are presented in Table 4. The model shows whether it supports the relationships established or not. The findings reveal that residents' place image significantly affects perceived socio-cultural impacts ($\beta$ = 0.640; t = 17.655; $p$ < 0.01) and perceived economic impacts ($\beta$ = 0.652; t = 17.850; $p$ < 0.01). Therefore, hypotheses H1 and H2 are confirmed.

**Table 4.** Hypotheses testing results.

| | | Original Sample (O) | Sample Mean (M) | Standard Deviation (STDEV) | T Statistics (|O/STDEV|) | *p* |
|---|---|---|---|---|---|---|
| H$_1$ supported | RPI → PSCI | 0.640 | 0.641 | 0.036 | 17.655 | 0.000 |
| H$_2$ supported | RPI → PECI | 0.652 | 0.653 | 0.037 | 17.850 | 0.000 |
| H$_3$ not supported | RPI → PENI | 0.172 | 0.171 | 0.102 | 1.693 | 0.091 |
| H$_4$ supported | PSCI → SUST | 0.263 | 0.268 | 0.073 | 3.585 | 0.000 |
| H$_5$ not supported | PECI → SUST | −0.157 | −0.156 | 0.073 | 2.150 | 0.032 |
| H$_6$ supported | PENI → SUST | 0.277 | 0.277 | 0.057 | 4.828 | 0.000 |

Notes: RPI = Residents' Place Image, PSCI = Perceived Socio-cultural Impacts; PENI = Perceived Environmental Impacts; PECI = Perceived Economic Impacts; SUST = Support for Sustainable Tourism.

On the other hand, the results revealed that the residents' place image was not a significant predictor of the perceived environmental impacts ($\beta$ = 0.172; t = 1.693; $p$ > 0.05). Therefore, H3 has been rejected. It has been determined that the perceived socio-cultural impacts variable has a significant effect on the support for sustainable tourism ($\beta$ = 0.263; t = 3.585; $p$ < 0.01), and the H4 hypothesis has been confirmed accordingly.

The analysis has shown that the perceived economic impacts variable has a negative effect on support for sustainable tourism ($\beta$ = −0.157; t = 2.150; $p$ < 0.05), which contradicts the H5 hypothesis. Therefore, the H5 hypothesis is rejected. On the other hand, the perceived environmental impacts variable has a positive effect on support for sustainable tourism ($\beta$ = 0.277; t = 4.828; $p$ < 0.01), confirming the H6 hypothesis. The t-values for the standardized ($\beta$) coefficients of the confirmed hypotheses are above 1.96 and at a 95% confidence level.

Furthermore, the $R^2$ values for the proposed model show that perceived socio-cultural impacts and perceived economic impacts explain 41% and 42% of the variance in support for sustainable tourism, respectively. This indicates that the proposed model explains a substantial amount of variance in support for sustainable tourism. The $R^2$ values obtained are much higher than the threshold value of 0.10, except for perceived environmental impacts, which explains only 3% of the variance in support for sustainable tourism and is, therefore, excluded [90].

## 5. Discussion

Residents are a crucial stakeholder in tourism development and are often considered the industry's backbone [74,91]. In order to promote sustainable tourism, it is important for tourism planners to involve the local population in the decision-making process [50,92]. Local residents should also be informed about the benefits of tourism development, while ensuring that all members of the community, including minorities and vulnerable groups, have an equal opportunity to benefit from tourism-related activities [76,77]. Midyat, where this study was conducted, is a significant tourism destination due to its historical structures and its diverse religious population, which has gained popularity in recent years. Understanding the city's image from the perspective of the residents can provide Midyat's tourism professionals with valuable insights for developing effective strategies in the future, making the destination's tourism more manageable and sustainable.

Overall, the study highlights the importance of residents' participation and involvement in tourism decision-making processes to ensure sustainable tourism development. The case study of Midyat demonstrates the significance of understanding the local residents' place image and its impact on the perceived socio-cultural, environmental, and economic impacts of tourism. The confirmed hypotheses in the study show that perceived

environmental impacts have a positive effect on support for sustainable tourism, while perceived economic impacts have a negative effect. This information can guide tourism planners and professionals in Midyat to make more informed decisions and take stronger steps toward sustainable tourism development.

In this study, transportation attributes, social attributes, government service attributes, and shopping have been identified as secondary-level hidden structures within the residents' place image, which is also used in a study by Ramkissoon and Nunkoo (2011) [37]. The hypotheses were tested using a measurement model and structural equation modeling, and it was found that the residents' place image has a positive effect on perceived socio-cultural and economic impacts. These findings are similar to those of Jurowski et al. (1997) [93] and Stylidis et al. (2014) [35]. However, it was not found that the residents' place image affects perceived environmental impacts, which is not consistent with the result obtained in the study conducted by Stylidis et al. (2014) [35]. A positive image of the city among residents can lead to greater support for tourism development in the destination. Therefore, it is suggested that tourism policy should aim to improve the residents' image of the city because they are more likely to support the development of tourism if they have a positive perception of the urban environment. This is in line with Ramkissoon and Nunkoo's (2011) [37] study. It has been found that the residents' perceived socio-cultural and environmental impacts increase support for sustainable tourism. This finding is consistent with the studies of Uslu et al. (2020) [36], Jurowski et al. (1997) [93], and Stylidis et al. (2014) [35]. However, it has been determined that the residents' perceived economic impacts do not have a positive effect on support for sustainable tourism. Contrary to the hypothesis formed, a different result has been obtained: the residents' perceived economic impacts decrease support for sustainable tourism. These results differ from the studies of Demirovic, Bajrami et al. (2020) [94], Vargas-Sánchez et al. (2009) [25], Nunkoo and Ramkissoon (2012) [78], Stylidis et al. (2014) [35], and Uslu et al. (2020) [36]. As identified by two studies [95,96] in the literature, residents' place image can significantly affect the rejection or acceptance of proposed tourism development projects. The reason why economic impacts do not have a positive effect on residents' support for sustainable tourism may be related to the source of their income. However, further research is required to explain this phenomenon, including the inclusion of questions regarding whether residents obtain income from tourism or not.

Moreover, the study found that the residents' life standard, job opportunities, infrastructure, and economic income are insufficient in Midyat, but the city is perceived as more friendly, trusting, and supportive in terms of social attributes. Residents with friendlier place images tend to have higher tolerance levels for development impacts, which is consistent with the findings in the tourism literature [40,97]. Therefore, all stakeholders involved in the tourism destination must undertake activities that will create a positive destination image and increase the perceived impacts of tourism in a positive direction. Residents need to be more involved in tourism activities by providing job opportunities, which would increase their income from tourism and improve their standard of living. The local government needs to work on enhancing the city's infrastructure. It is also important to engage residents in economic activities through their contributions. By implementing these suggestions, Midyat, as a tourism destination, would significantly improve in a sustainable manner, and residents' support would be enhanced. However, to transform Midyat's tourism activities into sustainable tourism and create a positive destination image in the minds of tourists and residents, all stakeholders must work together. As a consequence of the results, it is recommended that efforts should be made to improve the image of the city by enhancing government services, transportation infrastructure, and shopping amenities. It is also important to actively involve the residents in tourism development to provide them with income opportunities from tourism, as well as to diversify their sources of income as a sustainable strategy.

## 6. Conclusions and Limitations

Nowadays, the positive impacts of tourism, such as contributing to national income, creating foreign currency inflows, and generating employment opportunities, present themselves as large-scale economic sources of power for cities, destinations, and countries. This phenomenon, called the "smokeless industry", is an important factor for every city, destination, and country. Those tourism destinations that use their resources effectively and efficiently will have stronger positive impacts. At this juncture, it is undoubtedly possible to ensure the continuity of tourism resources and prevent the burdening of future generations' tourism needs through sustainable tourism.

The physical development of tourism in an area brings about many socio-cultural, economic, and environmental changes to the residents living in that area. Negative perceptions of tourism development by affected residents, and the development of negative attitudes in parallel, can lead to inconveniences that hamper tourism development and create a disturbing environment for tourists. Therefore, it is beneficial to understand the perceptions and attitudes of the residents towards tourism in order to promote healthy tourism development. Involvement of residents in tourism, if necessary, can raise awareness and promote sustainable tourism practices in cities. The urban community, with its richness of heritage and culture, plays a crucial role in advancing sustainable development through the tourism industry. However, if tourism is not properly managed, residents may withdraw their support for tourism development as it progresses.

This paper provides evidence that sustainable tourism development is crucial for a city's tourism attractiveness. To ensure the support of residents, it is necessary to respect their attitudes towards the impacts of tourism, including socio-cultural and environmental impacts. Based on the above, it can be concluded that strategic positioning, symbiotic partnerships, active community participation, and innovative governance can guarantee an inclusive and sustainable future for tourism. Lastly, sustainable urban tourism can have a positive impact on the image of the city and attract more visitors to the city.

This study has some limitations. The findings are limited to the attitudes of residents living in the Midyat district center in Turkey. Evaluating sustainable tourism development with only one tourism stakeholder is not sufficient. The participation of all tourism stakeholders is relevant. Therefore, further studies need to be conducted to ensure the contribution of other tourism stakeholders, not only residents.

**Author Contributions:** Conceptualization, A.U., E.E., J.A.C.S. and S.O.; methodology, A.U. and E.E.; software, A.U.; validation, E.E. and A.U.; formal analysis, A.U. and E.E.; investigation, A.U., E.E., J.A.C.S. and S.O.; resources A.U., E.E. and S.O.; data curation, A.U. and E.E.; writing—original draft preparation, E.E., J.A.C.S. and S.O.; visualization, A.U., E.E., J.A.C.S., S.O. and M.C.S.; writing—review and editing, A.U., E.E., J.A.C.S., S.O. and M.C.S.; supervision, A.U., E.E. and J.A.C.S. All authors have read and agreed to the published version of the manuscript.

**Funding:** This research received no external funding.

**Institutional Review Board Statement:** Not applicable.

**Informed Consent Statement:** Not applicable.

**Data Availability Statement:** Not applicable.

**Conflicts of Interest:** The authors declare no conflict of interest.

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
