# Peer review of "Determinants of Residents’ Support for Sustainable Tourism Development: An Empirical Study in Midyat, Turkey"

_sustainability, doi:10.3390/su151310013_

Round 1
Reviewer 1 Report
The article focus on interesting topic, the only elements needing correction are mentioned below.
The aspects of methods about how the questionnaires were collected and when should be explained.
The statement in lines 388-389 seems not to be proven within the article.
Small language mistaek to be corrected in Line 122 - bottom;
Author Response
Reviewer's comments:
The article focus on interesting topic, the only elements needing correction are mentioned below.
The aspects of methods about how the questionnaires were collected and when should be explained.
Authors response:
Thank you for your comments and suggestions. We feel that by addressing your concerns (and other reviewers) we have improved our work. Thank you for your time and insight in allowing us to improve and resubmit this work!
We agree with you. Hence, we added the following sentence. Below is the revised text (please see the methods section in page 5):
«The questionnaires were collected by using convenience sampling technique between May-August, 2020».
Reviewer comment: The statement in lines 388-389 seems not to be proven within the article.
Authors' response:
You are correct in stating this. Based on your recommendation we revised the sentence in lines 388-389. We deleted the economic impacts.
‘To ensure the support of residents, it is necessary to respect their attitudes towards the impacts of tourism, including socio-cultural, and environmental impacts’. (please see the page 11)
Reviewer comment:
Small language mistaek to be corrected in Line 122 - bottom.
(please see page 3).

Reviewer 2 Report
Dear Authors,
The content of the manuscript concerns an interesting and important research thread, especially since the inhabitants are the stakeholders of the local tourism market and often the creators of a diversified product.
I believe that the hypotheses were well formulated and the research procedure was properly applied.
The obtained results can still be compared with those obtained by the authors of the text: "Determinants of Residents’ Support for Sustainable Tourism Development: Implications for Rural Communities”. In Sustainability 2020, 12(22): 9438.
Author Response
Reviewer's comment:
Dear Authors,
The content of the manuscript concerns an interesting and important research thread, especially since the inhabitants are the stakeholders of the local tourism market and often the creators of a diversified product. I believe that the hypotheses were well formulated and the research procedure was properly applied.
Authors' reply:
We thank you for this wonderful compliment. It is our great honor to be considered for publication in Sustainability.
Furthermore, we thank you for taking the time to review our work and providing your insight to help improve our research. Thank you for your support and valuable comments.
Reviewer's comment:
The obtained results can still be compared with those obtained by the authors of the text: "Determinants of Residents’ Support for Sustainable Tourism Development: Implications for Rural Communities”. In Sustainability 2020, 12(22): 9438.!
Authors' reply:
We appreciate you pointing this out to us. We added the suggested source (i.e., Demirovic Bajrami et al., 2020) and compared with our studies.
Demirović Bajrami, D.; Radosavac, A.; Cimbaljević, M.; Tretiakova, T.N.; Syromiatnikova, Y.A. Determinants of Residents’ Support for Sustainable Tourism Development: Implications for Rural Communities. Sustainability 2020, 12, 9438. https://doi.org/10.3390/su12229438
(please see the page 10 and references list)

Reviewer 3 Report
The study could be better written up in terms of findings and recommendations could be better communicated.
As it is a sustainability for tourism specific to residents article it seems a good idea to provide significant factors, such as income, as relevant to other industries.
See comments. Abstract
Line 25-26: If it was in support or not.
Line 29: Recommendations include……… with limitations regarding study cohort recognised, only residents.
Line 338-240; 350-351 provide significant findings or recommendations. See comments about ordering the paragraphs.
Provide recommendations specific to finding regarding perceived economic impact, (accurate or not), and whether income is an associated factor, and if income needs to be from tourism.
Line 353-359: Write up of findings, and order, provide findings, then suggestions and recommendations.
Line 362-363: Classify this recommendations with study questions, citation or other.
Line 366-367: Findings depend on stratgey for midday as tourist destination. Affordability to visit and other factors which would attract different market and change competitors.
Must also consider sustainable enhancements for each of the three recommendations.
Include consideration of other income opportunities for residents, in an effort to not rely on the studied industry. As a sustainability point.
Line 383: Differentiate between active involvement in tourism, and involvement in tourist development.
Author Response
Reviewer's comment:
The study could be better written up in terms of findings and recommendations could be better communicated. As it is a sustainability for tourism specific to residents article it seems a good idea to provide significant factors, such as income, as relevant to other industries.
Authors' reply:
We were extremely thorough in addressing each of the following comments made by yourself and other reviewers. We believe that responding to your following comments most certainly improves the quality of our manuscript. We hope these changes will get your approval. Furthermore, we thank you for taking the time to review our work and providing your insight in ways to help advance our work.
Reviewer's comment:
See comments. Abstract, Line 25-26: If it was in support or not.
Authors' reply:
Thank you. We have detailed the study’s findings to make the abstract more complete and unsderstandable. We have added the following text: « While perceived environmental impacts have a positive effect on support for sustainable tourism, perceived economic impacts have a negative effect. This finding can guide tourism planners and professionals to make more informed decisions and take stronger steps toward sustainable tourism development.»
(please see the abstract)
Reviewer's comment:
Line 29: Recommendations include……… with limitations regarding study cohort recognized, only residents.
Authors' reply:
Thanks for your comment.
We have reformulated the abstract and deleted the study’s limitations, which should appear only at the end of the paper. There, we have detailed the limitations and recommendations for future studies.
Reviewer's comment:
«Line 338-240; 350-351 provide significant findings or recommendations. See comments about ordering the paragraphs.
Provide recommendations specific to finding regarding perceived economic impact, (accurate or not), and whether income is an associated factor, and if income needs to be from tourism.
Authors' reply:
Thank you for your comment.
We have provided recommendations and explained them. "The reason why economic impacts do not have a positive effect on residents' support for sustainable tourism may be related to the source of their income. However, further research is required to explain this phenomenon, including the inclusion of questions regarding whether residents obtain income from tourism or not."
(please see page 11)
Reviewer's comment:
Line 353-359: Write up of findings, and order, provide findings, then suggestions and recommendations.
Authors' reply:
We have added suggestions and recommendations based on the findings. “Residents need to be more involved in tourism activities by providing job opportunities, which would increase their income from tourism and improve their standard of living. The local government needs to work on enhancing the city's infrastructure. It is also important to engage residents in economic activities through their contributions. By implementing these suggestions, Midyat, as a tourism destination, would significantly improve in a sustainable manner, and residents' support would be enhanced”
(please see page 11)
Reviewer's comment:
Line 362-363: Classify this recommendations with study questions, citation or other.
Authors' reply:
Thanks for your comment. We agree with you as the recommendation on lines 362-363 «This includes forming a good destination image in the minds of domestic tourists through the film and series industry» seems a bit out of context. Therefore, we have deleted it. Indeed, the text has improved.
Reviewer's comment:
Line 366-367: Findings depend on stratgey for midday as tourist destination. Affordability to visit and other factors which would attract different market and change competitors.
Must also consider sustainable enhancements for each of the three recommendations.
Include consideration of other income opportunities for residents, in an effort to not rely on the studied industry. As a sustainability point.
Authors' reply:
We appreciate your remark and fully agree with you. While it is important that on destination level the residents should benefit directly or indirectly from tourism, it is a wise suggestion to diversify the economic activities and sources of income of residents as a sustainable strategy. Therefore, we have reformulated the sentence to include your valuable suggestion: «It is also important to actively involve the residents in tourism development to provide them with income opportunities from tourism as well as diversify their sources of income as a sustainable strategy»
Reviewer's comment:
Line 383: Differentiate between active involvement in tourism, and involvement in tourist development.:
Authors' reply:
Thank you for bringing this to our attention. Below is the revised paragraph of the conclusion and limitations:
We have modified it to “Involvement of residents in tourism”.

Reviewer 4 Report
This study aims to investigate the relationship between the place image of Midyat 163 residents and their perceptions of the impacts of tourism, specifically in terms of socio-164 cultural, economic, and environmental dimensions, and how this ultimately affects their 165 support for tourism development. (line 163-166) The objective concerns the development of tourism in general or sustainable tourism?
In the case of sustainable tourism, it is a slightly different development than in the case of conventional tourism. It seems to me that sustainable tourism and the factors for developing it are not emphasised enough in the description of the studies you have carried out: e.g. Table 2 line 254 it might be worth pointing out which indicators relate to sustainable tourism (the three indicated are not sufficient in my opinion, but others could also be selected from this table)
Author Response
Reviewer's comment:
This study aims to investigate the relationship between the place image of Midyat residents and their perceptions of the impacts of tourism, specifically in terms of sociocultural, economic, and environmental dimensions, and how this ultimately affects their support for tourism development. (line 163-166) The objective concerns the development of tourism in general or sustainable tourism?
In the case of sustainable tourism, it is a slightly different development than in the case of conventional tourism. It seems to me that sustainable tourism and the factors for developing it are not emphasised enough in the description of the studies you have carried out: e.g. Table 2 line 254 it might be worth pointing out which indicators relate to sustainable tourism (the three indicated are not sufficient in my opinion, but others could also be selected from this table).
Authors' answer:
Thank you for your comments and we will try to clarify your question. Of course, the tourism development that is intended to Midyat is sustainable tourism development, which can not be achieved without residents’ support for tourism. Therefore, studying residents’ perceptions of the impacts of tourism (in terms of socio-cultural, economic, and environmental dimensions) is crucial to determine their support for tourism and for the success of sustainable tourism development.
In this study we do not make a distinction between sustainable tourism and other forms of tourism which may be less sustainable. All forms of tourism can be and must be made more sustainable. But as said before, yes, the tourism type that is intended for Midyat is sustainable tourism development. Therefore, studying residents perceptions of tourism impacts considering the different dimensions of sustainability is crucial for understanding if tourism in Midyat is moving towards sustainable tourism development.
We hope we have clarified your question.
Many thanks!
